

# Dual mechanism of TRKB activation by anandamide through CB1 and TRPV1 receptors

Cassiano R.A.F. Diniz[1], Caroline Biojone[2,3], Samia R.L. Joca[3,4,5], Tomi Rantamäki[6], Eero Castrén[2], Francisco S. Guimarães[1] and Plinio C. Casarotto[1,2]

[1] Department of Pharmacology, Ribeirão Preto Medical School, University of São Paulo, São Paulo, Brazil
[2] Neuroscience Center—HILIFE, University of Helsinki, Helsinki, Finland
[3] Department of Physics and Chemistry, Ribeirão Preto School of Pharmaceutical Sciences, University of São Paulo, Ribeirão Preto, São Paulo, Brazil
[4] Department of Clinical Medicine, Translational Neuropsychiatric Unit, Aarhus University, Aarhus, Denmark
[5] Aarhus Institute of Advanced Studies, Aarhus University, Aarhus, Denmark
[6] Division of Pharmacology and Pharmacotherapeutics, Faculty of Pharmacy, University of Helsinki, Helsinki, Finland

Corresponding author
Plinio C. Casarotto,
plinio.casarotto@helsinki.fi

## ABSTRACT

**Background.** Administration of anandamide (AEA) or 2-arachidonoylglycerol (2AG) induces CB1 coupling and activation of TRKB receptors, regulating the neuronal migration and maturation in the developing cortex. However, at higher concentrations AEA also engages vanilloid receptor TRPV1, usually with opposed consequences on behavior.

**Methods and Results.** Using primary cell cultures from the cortex of rat embryos (E18) we determined the effects of AEA on phosphorylated TRKB (pTRK). We observed that AEA (at 100 and 200 nM) induced a significant increase in pTRK levels. Such effect of AEA at 100 nM was blocked by pretreatment with the CB1 antagonist AM251 (200 nM) and, at the higher concentration of 200 nM by the TRPV1 antagonist capsazepine (200 nM), but mildly attenuated by AM251. Interestingly, the effect of AEA or capsaicin (a TRPV1 agonist, also at 200 nM) on pTRK was blocked by TRKB.Fc (a soluble form of TRKB able to bind BDNF) or capsazepine, suggesting a mechanism dependent on BDNF release. Using the marble-burying test (MBT) in mice, we observed that the local administration of ACEA (a CB1 agonist) into the prelimbic region of prefrontal cortex (PL-PFC) was sufficient to reduce the burying behavior, while capsaicin or BDNF exerted the opposite effect, increasing the number of buried marbles. In addition, both ACEA and capsaicin effects were blocked by previous administration of k252a (an antagonist of TRK receptors) into PL-PFC. The effect of systemically injected CB1 agonist WIN55,212-2 was blocked by previous administration of k252a. We also observed a partial colocalization of CB1/TRPV1/TRKB in the PL-PFC, and the localization of TRPV1 in CaMK2+ cells.

**Conclusion.** Taken together, our data indicate that anandamide engages a coordinated activation of TRKB, via CB1 and TRPV1. Thus, acting upon CB1 and TRPV1, AEA could regulate the TRKB-dependent plasticity in both pre- and postsynaptic compartments.

## INTRODUCTION

Repetitive behaviors are part of the normal background of animal's learning (*Langen et al., 2011b*; *Langen et al., 2011a*). These stereotypic motor procedures are regulated by the cortical-striatal-thalamo-cortical (CSTC) circuitry, a series of multiple reverberatory loops, with the prefrontal cortex, striatum and thalamic nuclei playing a core role as integrative hubs (*Alexander, DeLong & Strick, 1986*). Changes in the CSTC can trigger abnormal repetitive behaviors. These aberrant behaviors can be related to several human neuropsychiatric conditions such as Huntington's disease and obsessive-compulsive disorder (OCD). In a simplified version, the CSTC is divided into two major pathways: direct and indirect (*Casarotto, Gomes & Guimarães, 2015*). The direct pathway comprises the cortical projections to *globus pallidus* (*pars internalis*) and *substantia nigra pars reticulata,* while the indirect pathway engages a more complex set of relay structures, involving the *globus pallidus pars externalis* and subthalamic nucleus (for review see *Canales & Graybiel, 2000*; *Canales & Graybiel, 2000*). The resultant activity between these two pathways regulates the output of the basal ganglia in favor of one of the two possible effects: increase in the repetitive movements (favored by the direct pathway activity) or inhibition of such programs, a consequence of activation of the indirect pathway. Although highly simplified, this model of the CSTC circuit provides a useful framework for understanding circuit physiology and putative dysfunctions (*Casarotto, Gomes & Guimarães, 2015*).

Multiple neurotransmitters/neuromodulators systems act in coordination to regulate the balance in the CSTC circuitry. Among them, endocannabinoids play a central role regulating not only glutamatergic, but also GABAergic, serotonergic, and dopaminergic transmission (for review see (*López-Moreno et al., 2008*; *López-Moreno et al., 2008*)). Briefly, the activation of CB1 receptors by AEA (N-arachidonoylethanolamine) or 2AG (2-arachidonoylglycerol), produced as 'on-demand retrograde messengers', usually decreases the activity of presynaptic neurons via Gi/0 and modulation of calcium and potassium channels (for review see (*Chevaleyre, Takahashi & Castillo, 2006*; *Chevaleyre, Takahashi & Castillo, 2006*)). Due to these effects, endocannabinoids are putatively able to regulate excessive neurotransmission in the CSTC system. Otherwise, endocannabinoids are also described to trigger Gq downstream signaling at astrocytes to increase calcium intracellular levels, and others endocannabinoids such as N-arachidonoyl-dopamine are even potent agonists to TRPV1 (*Hashimotodani et al., 2007*; *Castillo et al., 2012*). Besides, the synthesis and release of endocannabinoids usually is triggered by depolarization-induced calcium influx, as well as by activated phospholipase-C-beta following activation of Gq-protein coupled receptors (*Hashimotodani et al., 2007*; *Castillo et al., 2012*).

Endocannabinoids can also act on TRPV1 receptors. These receptors increase calcium influx, facilitating a rapid depolarization of the neuronal cells (*Casarotto, de Bortoli & Zangrossi Jr, 2012*; *Moreira et al., 2012*). In preclinical anxiety models, high doses of AEA are usually ineffective or cause anxiogenic rather than anxiolytic effects. This bell-shaped dose–effect curve has been associated with TRPV1 activation and is reversed by pretreatment with antagonists of these receptors (*Casarotto, de Bortoli & Zangrossi Jr, 2012*; and for review see

*Moreira & Wotjak, 2010*; *Aguiar et al., 2014*). Accordingly, the anxiolytic effect of capsaicin, an agonist of TRPV1 receptors, observed after intracerebral administration was attributed to a desensitization of the channels (*Terzian et al., 2009*).

CB1 receptors are highly expressed in the anterior cingulate cortex, striatum and *substantia nigra* (*Harkany et al., 2007*; *Díaz-Alonso, Guzmán & Galve-Roperh, 2012*), major hubs of CSTC circuitry. TRPV1 has a less broad expression when compared to CB1 (*Tóth et al., 2005*; *Menigoz & Boudes, 2011*). However, these two receptors are colocalized in the periaqueductal grey matter (PAG) and prefrontal cortex playing opposite functional roles (*Casarotto, de Bortoli & Zangrossi Jr, 2012*; *Fogaça et al., 2012*).

CB1 receptors can couple to and transactivate tyrosine kinase receptors (*Dalton & Howlett, 2012*). Indeed, TRKB (tropomyosin-related kinase B) has been colocalized with CB1 receptors (*Berghuis et al., 2005*). These authors suggested that AEA leads to coupling between TRKB and CB1. TRKB is the receptor of brain-derived neurotrophic factor (BDNF), a member of the neurotrophic family that includes NGF, NT-3 and NT4/5 (*Bothwell, 2014*). The activation of TRKB is a crucial factor in the regulation of brain plasticity (*Castrén & Antila, 2017*). Animals lacking CB1 receptors (CB1.KO) display behavioral despair and a reduction in BDNF levels in hippocampus (*Aso et al., 2011*), suggesting an interplay between these two systems during development and in adulthood. CB1.KO animals also exhibit impaired extinction of aversive memories (*Marsicano et al., 2002*), a feature usually facilitated by the activation of TRKB following antidepressant treatment, and prevented by reduced BDNF expression (*Karpova et al., 2011*).

Thus, the initial goal of the present study was to investigate if CB1 and TRKB systems interact in a mouse model of anxiety/repetitive behavior, aiming to replicate the *in vitro* mechanism proposed by Berghuis and colleagues (*Berghuis et al., 2005*). Following the results obtained in the present study, we then speculate on the participation of TRPV1 in the AEA effects on TRKB.

## MATERIAL AND METHODS

### Animals
Male and female C57BL6/j were used (weight 25–30 g, 12–18 weeks old at the start of the experiments), with free access to food and water, except during experimental procedures. All procedures were approved by the University of São Paulo (protocol: 146/2009) and for the experiments conducted at the University of Helsinki (protocol: ESAVI/10300/04.10.07/2016), in accordance with international guidelines for animal experimentation.

### Drugs
Anandamide (AEA, #1017; Tocris, Bethesda, MD, USA), arachidonyl-2′-chloroethylamide (ACEA, #1319, Tocris, USA), WIN 55,212-2 mesylate (WIN, #1038; Tocris), capsaicin (CPS, #0462; Tocris, USA), AM251 (#1117; Tocris), TRKB.Fc (#688-TK-100; R&D Systems, Minneapolis, MN, USA), capsazepine (CZP, #0464; Tocris), k252a (#05288, Sigma-Aldrich, USA) and recombinant-human brain-derived neurotrophic factor (rhBDNF, #450-02; Peprotech, Rocky Hill, NJ, USA) were used. AEA and ACEA

were dissolved in Tocrisolve (#1684; Tocris). WIN and AM251 were suspended in 2%Tween 80, and k252a was diluted in DMSO 0.2%, and dissolved in sterile saline for ip injections. CPS, CZP or AM251 were diluted in DMSO (at 1,000×the final concentration used in the assays) for *in vitro* experiments. BDNF and TRKB.Fc were dissolved in phosphate buffered saline (PBS). The doses of the drugs used to intracerebral infusions were based on previous works which have shown those doses being effective to change the animal behavior (*Casarotto et al., 2010*; *Casarotto et al., 2012*; *Casarotto, de Bortoli & Zangrossi Jr, 2012*; *Fogaça et al., 2012*; *Terzian et al., 2014*).

### Primary cultures

Cortex of E18 rat embryos were dissected and the cells plated in poly-L-lysine coated wells at a $0.5 \times 10^6$ cells/ml density in Neurobasal medium (*Rantamäki et al., 2011*; *Sahu et al., 2018*). The cells were left undisturbed, except for medium change, for 8 days before drug treatment and sample collection.

## APPARATUS AND PROCEDURE

### Marble burying test (MBT)

Test was performed using a clear square box ($30 \times 20 \times 12$ cm) with 5 cm sawdust layer covered floor. Twelve green clear glass marbles (1.5 cm in diameter) were evenly spaced over the floor. One hour before test the animals were left undisturbed in the experimental room. Mice were initially placed on the box's center containing marbles. Twenty-five min later they were taken from the box, and the number of buried marbles was counted. Criteria for buried marbles included only those with at least two-thirds under sawdust (*Njung'e & Handley, 1991*).

### Locomotor activity

Male mice were submitted to a 25 min session in a circular arena (40 cm in diameter with a 50 cm high plexiglass wall). Each session was videotaped, and the total activity (distance travelled by the mice in meters) was analyzed with the help of the ANY-maze software (version 4.5; Anymaze, Stoelting, Dublin, Ireland). The apparatus was previously cleaned with 70% alcohol solution before each animal assessment.

### Stereotaxic surgery

Briefly, male mice were anesthetized with tribromoethanol and fixed in a stereotaxic frame. Next, stainless steel guide cannulas (0.7 mm OD, 9 mm length) were implanted into the prelimbic ventromedial prefrontal cortex (PL; coordinates: $AP = -2.8$ mm from bregma, $L = 1.5$ mm, $DV = 2$ mm; lateral inclination of 20°) and then attached to skull bone with steel screws and acrylic cement. Obstruction was prevented with a stylet inside guide cannula. To avoid post-operative complications and any subsequent malaise, animals received during the surgery a subcutaneous dose of the anti-inflammatory banamine (0.25%, 0.1 ml/100 g; Schering-Plough, Kenilworth, NJ, USA) and a intramuscular dose of oxytetracycline (20%, 0.1 ml/100 g; Pfizer, New York, NY, USA). Soon after the surgery, the animals remained under surveillance in a heated and illuminated box until recovery from anesthesia. About five days after surgical procedure, bilateral intracerebral infusions in a

volume of 200 nl were performed with a dental needle (0.3 mm OD) during 1min. Hamilton microsyringe (Sigma-Aldrich, St. Louis, MO, USA) and an infusion pump (KD Scientific, Hollistan, MA, USA) were used. Following the behavioral tests, mice were anesthetized with chloral hydrate and 200nl of methylene blue was infused 1.5 mm deeper than the guide cannulas end. Brains were withdrawn and the injection sites verified. Data obtained from injections outside the aimed area were discarded. Coordinates for stereotaxic surgery and the histological location of the injections are based on the Paxinos' mouse brain atlas (*Paxinos & Franklin, 2001*).

## Sample collection, immunoprecipitation, and western blotting

Cell cultures at DIV8 were washed with ice-cold PBS and lysed using the following buffer (20 mM Tris-HCl; 137 mM NaCl; 10% glycerol; 0.05 M NaF; 1% NP-40; 0.05 mM Na3VO4) containing a cocktail of protease and phosphatase inhibitors (#P2714 and #P0044, respectively; Sigma Aldrich). The samples were centrifuged ($16,000 \times$ g) at 4 °C for 15 min and the supernatant was collected and stored $-80$ °C until use.

For the immunoprecipitation procedure, samples were incubated overnight at 4 °C with anti-TRKB (#AF1494, R&D systems, Abingdon, United Kingdom), anti-CB1 (#CB1-Rb-Af380; Frontier Institute, Ishikari, Hokkaido, Japan) or anti-TRPV1 (#sc12498; Santa Cruz Biotechology, Santa Cruz, CA, USA) at 1µg of antibody / 500 µg of total protein ratio or no antibody (negative control). Following incubation with 30 ul of Protein G-sepharose (#101242; Life Technologies, Carlsbad, CA, USA) the samples were centrifuged ($1,000 \times$ g/3min) and the supernatant discarded. Precipitated was washed 3 times with lysis buffer and stored at $-80$ °C until use. Samples were then added to the sample buffer, heated to 80 °C for 15 min, centrifuged at $1,000 \times$ g for 3 min and the supernatant resolved by SDS-PAGE, and finally transferred to PVDF membranes. The membranes were blocked with 3%BSA in TBST buffer (20 mM Tris-HCl; 150 mM NaCl; 0.1% Tween-20; pH 7.6) and incubated with anti-TRPV1 for 48 h at 4 °C. The detection was performed using HRP-conjugated rabbit anti-goat IgG (1:5,000, #2768, Santa Cruz, USA), followed by incubation with ECL and exposure to a CCD camera.

## Determination of pTRK levels

Levels of phosphorylated TRK were determined by ELISA (*Antila et al., 2014*). Briefly, 120 µg of total protein content from each sample was incubated in white 96-well plates previously coated with anti-TRKB (rabbit, 1:500, #sc-8316; Santa Cruz Biotechnology, Santa Cruz, CA, USA in carbonate buffer; pH 9.6) and blocked with 3% BSA in PBST buffer supplemented with Na3VO4 [0.1% Tween-80 in PBS; pH 7.6]. Following incubation with biotin-conjugated anti-phosphotyrosine (mouse, 1:2,000, #MCA2472B; AbD Serotec, Hercules, CA, USA) and streptavidin-HRP conjugate (1:10,000, #21126; Thermo Scientific, Waltham, MA, USA) the amount of luminescence was determined by incubation with ECL using a plate reader (Varioskan Flash; Thermo-Fisher, Waltham, MA, USA). The signal obtained from each sample was discounted of blank and normalized by the average of the control groups, thus expressed as percentage from control.

## Co-localization of CB1, TRPV1 and TRKB in the prefrontal cortex

Three female C57BL6/j mice were deeply anesthetized and perfused with saline followed by 4% PFA in PBS. The brains were removed from skull and post-fixated in 4% PFA at 4 °C for 48 h, and then cryoprotected in 30% sucrose. Coronal sections (40μm thick) were obtained in a vibratome (Leica VT1000-S). For immunostaining, the slices were washed in PBS, blocked for 30min at room temperature (3% BSA, 10% donkey normal serum in PBST containing 0.04% sodium azide), followed by incubation for 72 h at 4 °C with the primary antibodies, as follows: anti-CB1 (1:1,000), anti-TRPV1 (1:100, #sc12498 or #sc398417) and anti-TRKB (1:1,000). Another set of samples were labelled for CB1 or TRPV1, and CaMK2 (1:100, #AB22609; Abcam, Cambridge, UK). Additionally, a negative control staining without the primary antibody was included. After washing, sections were incubated 1 h at room temperature with 1:200 fluorophore-conjugated secondary antibodies (AlexaFluor donkey anti-goat, 647 nm #A21447; anti-rabbit, 568 nm #A11077; anti-mouse, 488 nm #A21202; Life Technologies, Carlsbad, CA, USA). Prelimbic prefrontal cortex was identified based on mouse brain atlas of Paxinos and Franklin (*Paxinos & Franklin, 2001*). Images were acquired at Biomedicum Imaging Unit at the University of Helsinki using a laser scanning confocal microscope (Zeiss LSM 780) (*Umemori et al., 2015*) equipped with 63×objective lens (Plan-Apochromat 63/1.40 oil DIC M27) and imaging Software ZEN (Zeiss, Oberkochen, Germany). A z-stack consisting of at least 8 consecutive images was obtained from each brain section, and images were later analyzed in ImageJ Software (NIH) (*Schneider, Rasband & Eliceiri, 2012*).

# EXPERIMENTAL DESIGN

## Concentration-response effects of AEA, CPS, and BDNF on TRKB phosphorylation and co-immunoprecipitation

Cortical cultures (DIV8) were treated with AEA (CB1/TRPV1 agonist; 0, 2.5, 10, 50, 100 or 200 nM), CPS (TRPV1 agonist; 0, 10, 100 or 200 nM) or BDNF (0, 5, 20 or 50 ng/ml, as positive control). Thirty min after drug administration the level of pTRK was determined by ELISA as described ($n = 4$-6/group). Next, lysates from cortical cells were pre-incubated with anti-CB1, anti-TRPV1 or anti-TRKB, and the levels of TRPV1 determined in the precipitated by western-blotting.

## CB1/TRPV1-dependent mechanism of AEA-activated TRKB

Cortical cells received AM251 (CB1 antagonist; 0 or 200 nM) followed 10min later by AEA (0 or 100 nM). Another batch of cells received AM251 (0 or 200 nM), followed 10 min later by BDNF (0 or 20 ng/ml). Next, cells were incubated with CZP (TRPV1 antagonist; 0 or 200 nM) followed 10min later by AEA (0, 100 or 200 nM). Concerning each experimental set above described, cells were lysed 30 min after the last drug administration for pTRK ELISA assay ($n = 5$-6/group).

## TRPV1-dependent mechanism of BDNF release by AEA

In another experimental set, cortical cells received TRKB.Fc (200 ng/ml) followed, 10min later, by CPS (0 or 200 nM), AEA (0 or 200 nM) or BDNF (0 or 20 ng/ml, used only as positive control but not part of the statistical analysis). Next, cortical cells received AM251

(CB1 antagonist; 0 or 200 nM) followed 10 min later by AEA (0 or 200 nM). Cells were lysed 30 min after the last drug administration for pTRK ELISA assay ($n = 5$-6/group).

## Effect of systemic and PFC local drug administration

Male mice received an ip injection of K252a (TRK antagonist; 0 or 80µg/kg) followed, 10min later, by another ip injection of WIN (CB1 agonist; 0 or 1 mg/kg). MBT test was performed 30min after the last injection ($n = 8$/group). Next, mice with implanted cannulae aimed at the prefrontal cortex received a bilateral injection of ACEA (CB1 agonist; 0.05 pmol), ACEA+k252a (10 pmol), CPS (5 nmol), CPS+k252a, BDNF (200 pg) or vehicle (DMSO) and were submitted to the MBT 20min later ($n = 5$-15/group). An independent group of animals received the same pharmacological treatments and was subjected to the locomotor activity test ($n = 5$-9/group).

## Statistical analysis

Data was analyzed by one- or two-way ANOVA with drug treatments as factors followed by Fisher's LSD *post-hoc* test when appropriate. Values of $p < 0.05$ were considered significant. All the data used in the statistical analysis, as well as background controls and low magnification images for immunostaining, in the present study is publicly available at figshare under CC-BY license (*Casarotto, 2018*).

# RESULTS

## *In vitro* concentration-response effects of AEA, CPS, and BDNF on TRKB phosphorylation

In order to verify whether CB1 and TRPV1 agonists would be able to activate TRK, AEA or CPS were infused into the medium of cultured cortical cells to then quantify the levels of phosphorylated TRK from the homogenized tissue. As it can be observed in Figs. 1A and 1B, both drugs were able to increase the levels of activated TRK. One-way ANOVA indicates a significant effect of AEA treatment in pTRK levels [$F(5,18) = 16.53$, $p < 0.0001$; Fig. 1A]. *Post-hoc* analysis indicates a difference between the vehicle- and AEA 100nM- and 200nM-treated groups (Fisher, $p = 0.0003$ and $<0.0001$, respectively). Administration of CPS also increased the levels of pTRK [$F(3,19) = 9.28$, $p = 0.0005$]. *Post-hoc* analysis indicates a difference between vehicle- and CPS 200 nM-treated groups (Fisher, $p < 0.0001$; Fig. 1B). Treatment with BDNF, used as a positive control, was also able to increase the levels of pTRK [$F(3,16) = 134.40$, $p < 0.0001$], with difference between vehicle- and 20 ng/ml- or 50ng/ml-treated groups (Fisher, $p < 0.0001$ for both; Fig. 1C). As seen in Fig. 1D, TRPV1 (VR1: band at 100 kDa) was detected in samples from primary cultures after precipitation of CB1 or TRKB, the ratio of CB1- and TRKB-coupled TRPV1 compared to total TRPV1 were: 27.0% and 13.2%, respectively. Thus, TRPV1 is suggested to be physically connected to the CB1 and TRKB.

## CB1/TRPV1-dependent mechanism of AEA-activated TRKB

As a next step, again using cultured cortical cells, we evaluated if the profile of TRK activation obtained with lower or higher concentrations of AEA depends on different mechanisms of action regarding CB1 and TRPV1 signaling as possible respective targets.

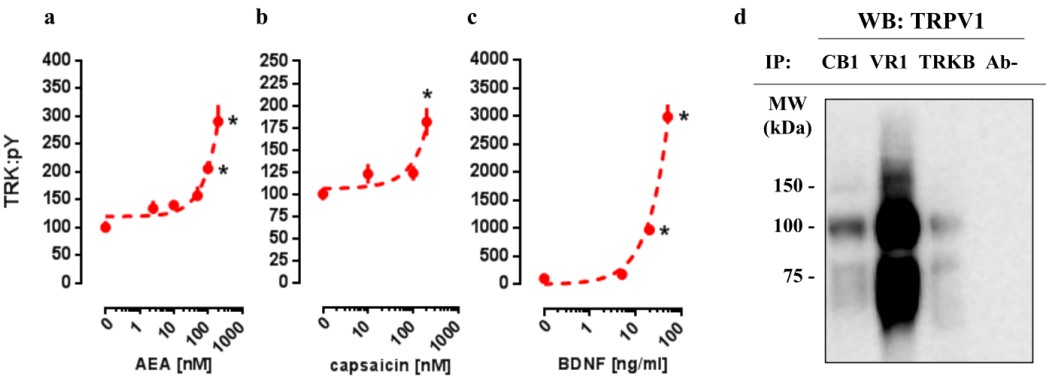

**Figure 1** *In vitro* concentration response effects of anandamide (AEA), capsaicin (CPS) and BDNF on TRKB phosphorylation. Cortical cultures were treated with (A) AEA 0, 2.5, 10, 50, 100 or 200 nM, (B) CPS 0, 10, 100 or 200 nM and (C) BDNF 0, 5, 20 or 50 ng/ml and 30 min after the pTRK levels of cell homogenate was measured ($n = 4 - 6$/group). Data is expressed as mean/SEM of percent from control group (0). Dashed lines represent the non-linear regression curve for the dose-response effect. * $p < 0.05$ compared to veh (0) group. (D) coIP of CB1 or TRKB and TRPV1 (VR1) in primary cultures of cortical cells.

First of all, effect of lower dose of AEA (100 nM) was checked on CB1-dependence, as previously infused AM251 abrogated TRK activation. Thus, two-way ANOVA indicates a significant interaction between AM251 and AEA effects on pTRK levels [$F(1,20) = 16.01$, $p = 0.0007$; Fig. 2A], with *post-hoc* analysis indicating a difference between veh/AEA100nM- and AM251/AEA100nM-treated groups (Fisher, $p < 0.0001$). The same interaction was not observed between AM251 and BDNF on pTRK levels [$F(1,16) = 0.85$, $p = 0.37$; Fig. 2B]. Next, two-way ANOVA indicates a significant interaction between a higher concentration of AEA (200 nM) and CZP on pTRK levels [$F(2,30) = 17.04$, $p < 0.0001$; Fig. 2C] while *post-hoc* test indicates a significant difference between the veh/AEA200 nM- and CZP/AEA200 nM-treated groups, but not between veh/AEA100 nM- and CZP/AEA100 nM-treated groups (Fisher, $p < 0.0001$). Thus, the higher concentration of AEA (200 nM) apparently increase even more the activation of TRK through TRPV1 activation.

## TRPV1-dependent mechanism of AEA-released BDNF

Cultured cortical cells were used to verify whether phopsphorylation of TRK from TRPV1 activation depend on BDNF release. In this case, two-way ANOVA indicates a significant interaction between the BDNF scavenger TRKB.Fc and the higher concentration of AEA (200 nM) treatment on pTRK levels [$F(2,18) = 15.01$, $p < 0.0001$; Fig. 2D], while *post-hoc* analysis indicates difference between veh/veh- and veh/AEA-treated groups (Fisher, $p < 0.0001$), but only a marginal difference between TRKB.Fc/veh- and TRKB.Fc/AEA-treated groups (Fisher, $p = 0.065$). This data suggest that the excess of TRK activation from AEA (200 nM) depend on TRPV1 activation. Next, previous data were corroborated inasmuch as two-way ANOVA depicts an interaction between the TRPV1 agonist CPS and TRKB.Fc treatments [$F(1,20) = 9.28$, $p = 0.0064$; Fig. 2E], and *post-hoc* test indicates a difference between veh/veh- and veh/CPS-treated groups (Fisher, $p = 0.0001$). Finally, effect of the higher dose of AEA (200 nM) partially depend on CB1 activation inasmuch as previously infused AM251 partially decreased pTRK levels. Thus, two-way ANOVA

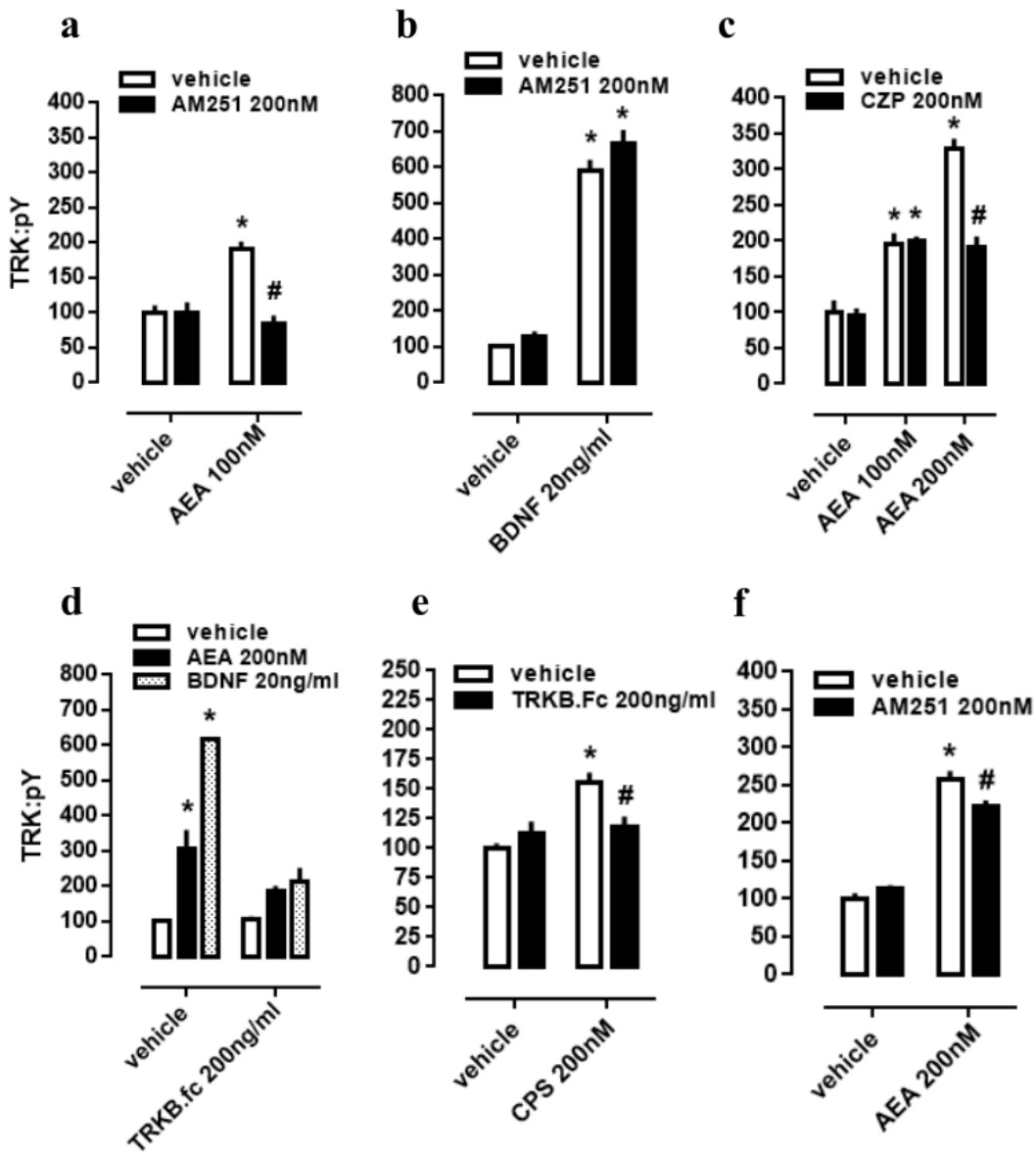

**Figure 2** **CB1/TRPV1-dependent mechanism of AEA-induced activation of TRKB.** Cortical cultures received AM251 (0 or 200 nM) followed 10 min later by (A) AEA (0 or 100 nM) or (B) BDNF (0 or 20 ng/ml). Cortical cells were incubated with (C) CZP (0 or 200 nM) followed 10 min later by AEA (0, 100 or 200 nM). Cortical cultures received (D) TRKB.Fc (0 or 200 ng/ml) followed 10 min later by (D) AEA (200 nM), BDNF (20 ng/ml); or (E) by CPS (200 nM). (F) Cortical cultures received AM251 (0 or 200 nM) followed 10min later by AEA (0 or 200 nM) ($n = 6$/group). Cells were lysed and homogenized 30 min after the last drug administration for pTRK measure ($n = 5 - 6$/group). Data is expressed as mean/SEM of percent from control group (veh-veh). * $p < 0.05$ compared to veh/veh group. # $p < 0.05$ compared to the respective veh/drug group.

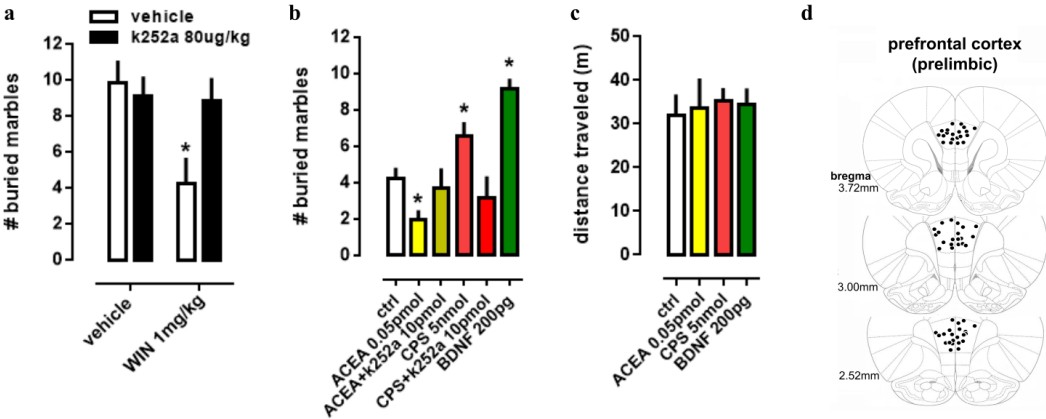

**Figure 3 Effect of systemic and PFC local drug administration.** (A) Mice received ip injection of K252a (80 ug/kg) followed, 10 min later, by ip injection of WIN (1 mg/kg) and were submitted to the MBT 30 min after the last drug injection ($n = 8$/group). (B) Mice implanted with a guide cannula aiming the prefrontal cortex received bilateral infusion of ACEA 0.05 pmol, ACEA+k252a, CPS 5 nmol, CPS+k252a or BDNF 200 pg, and were submitted to the MBT 20 min after drug administration ($n = 5 - 15$/group). Data is expressed as mean/SEM. (C) An independent cohort of animals received the same pharmacological treatment and was submitted to the locomotor activity test ($n = 5 - 9$/group). Data are expressed as mean ±SEM of distance traveled in meters. (D) Infusions are represented by points in the Paxinos's diagrams (2012). The number of points are fewer than the number of animals used due to overlap. * $p < 0.05$ compared to veh-treated (ctrl) group.

indicates a significant interaction between AM251 and AEA effects on pTRK levels [$F_{(1,20)}$ = 9.084, $p = 0.0069$; Fig. 2F], with *post-hoc* analysis indicating a difference between veh/AEA200nM- and AM251/AEA200nM-treated groups (Fisher, $p = 0.0048$).

## Effect of systemic and PFC local drug administration

In order to evaluate the interaction between TRK, CB1 and TRPV1 in a more complex way, a behavioral animal model was used to comprehend the actions of drugs infused into PL-PFC or injected intraperitoneally. Firstly, in a systemic approach, two-way ANOVA indicates a significant interaction between k252a and WIN treatments on marble burying behavior [$F_{(1,28)} = 4.52$; $p = 0.0425$; Fig. 3A], and *post-hoc* analysis depicted the pretreatment with k252a preventing the WIN-induced decrease in the number of buried marbles (veh/WIN vs k252a/WIN, Fisher, $p = 0.0125$). Now with intracerebral infusions, one-way ANOVA indicates a significant effect of drugs treatment [$F_{(5,53)} = 10.32$; $p < 0.0001$; Fig. 3B] with *post-hoc* test indicating that ACEA was able to decrease (Fisher, $p = 0.0095$), while CPS (5 nmol) and BDNF (200 pg) increased (Fisher, $p = 0.0253$ and $= 0.0003$, respectively) the number of buried marbles. Such effect of ACEA and CPS were counteracted by previous administration of k252a (Fisher, $p = 0.63$ and $= 0.37$, respectively). Thus, activation of both CB1 and TRPV1 may supposedly be acting through TRK activation to induce the opposite behavioral effect. No effect of drugs was observed on locomotor activity, as sees in Fig. 3C [$F_{(4,23)} = 0.11$; $p = 0.97$]. Then, locomotor parameters could not be a confounding factor to the behavioral data. Figure 3D depicts the injection sites in the PL-PFC of mice according to Paxinos' atlas (*Paxinos & Franklin, 2001*).

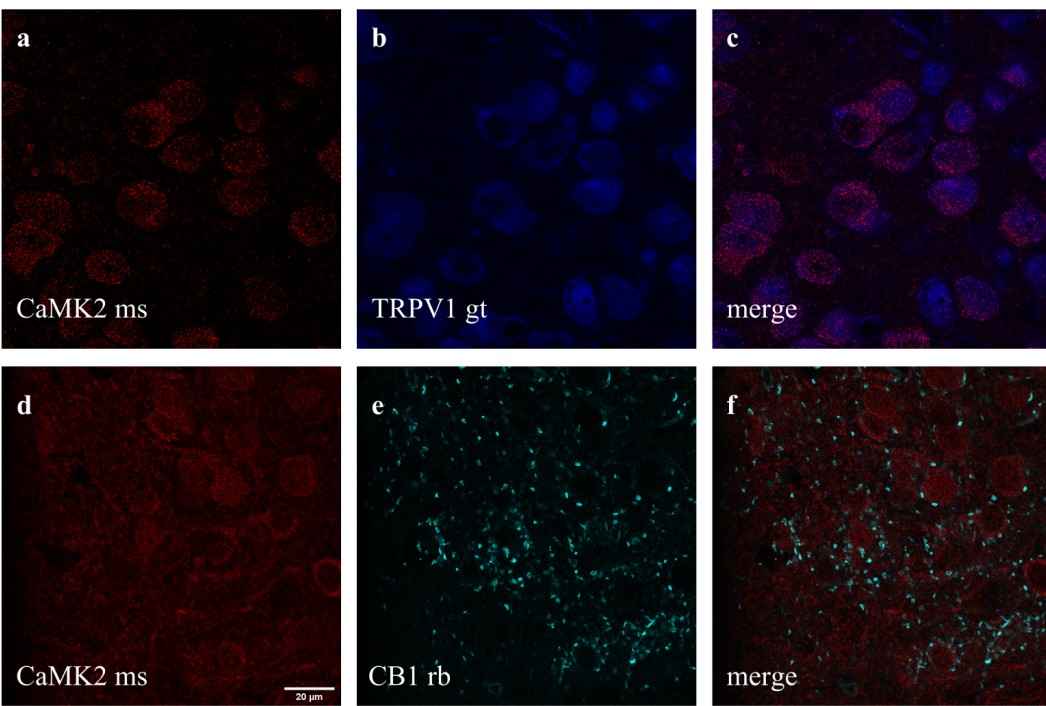

**Figure 4   Co-staining of (A–C) CaMK2 (from mouse - ms) and TRPV1 (from goat - gt), and (D–F) CaMK2 and CB1 (from rabbit - rb) in prelimbic area of prefrontal cortex in mice.** (A) cells positive to CaMK2 (red); (B) cells positive to TRPV1 (blue); (C) cells merged to both CaMK2 and TRPV1. (D) cells positive to CaMK2 (red); (E) cells positive to CB1 (cyan); (F) cells merged to both CaMK2 and CB1. Scale bar: 20 µm.

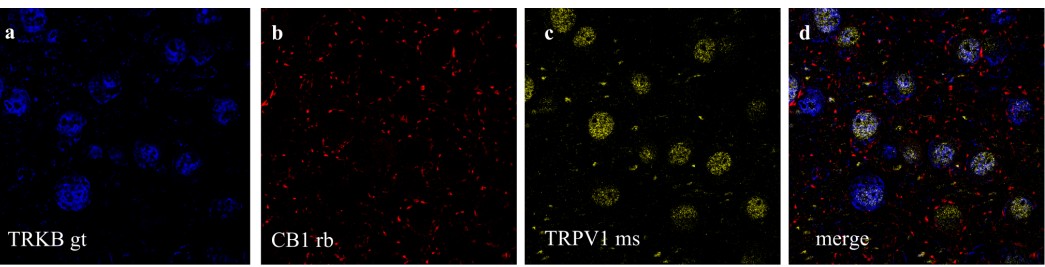

**Figure 5   Co-staining of (a) TRKB (from goat - gt), (b) CB1 and (c) TRPV1 (from mouse - ms) in pre-limbic area of prefrontal cortex in mice.** (A) cells positive to TRKB (blue); (B) cells positive to CB1 (red); (C) cells positive to TRPV1 (yellow); (D) cells merged to TRKB, CB1 and TRPV1. For scale bar please refer to Fig. 4D.

## Immunolabeling of CB1, TRPV1, and CaMK2 or TRKB

Double and triple immunofluorescence labeling were used to verify neuronal background in which TRPV1 and TRK are found, and to confirm CB1 presynaptic projection to excitatory neurons. As observed in Figs. 4A–4C, the majority of TRPV1+cells are also CaMK2+, while CB1 positive labels surround CaMK2+cells, Figs. 4D–4F. Figure 5, depicts

the triple labeling of TRPV, CB1 and TRKB. Again, the majority of TRPV1+cells were also positive for TRKB.

## DISCUSSION

In the present study we described a coordinated activation of TRKB (pTRK) by AEA in a CB1- and TRPV1-dependent manner. Lower concentrations of AEA (up to 100 nM) administered to cultured cortical cells were able to increase pTRK levels through CB1 activation. On the other hand, our pharmacological approach has led us to suggest that BDNF effect on pTRK levels does not engage endocannabinoids production, at least in the timeframe used in the present study. Higher concentration of AEA (200 nM) magnified the pTRK levels in a TRPV1-dependent manner, an effect mimicked by capsaicin (CPS). Both AEA (at 200 nM) and CPS required BDNF release to increase pTRK levels. However, CB1 and TRPV1 agonists, infused into PL-PFC produced opposite effects on burying behavior. CB1 agonist ACEA decreased, and TRPV1 agonist CPS increased the number of buried marbles. Further, BDNF infused into PL-PFC mimicked the behavioral effect of TRPV1 agonist. Data from total distance traveled indicates that the effects observed in the burying behavior could not be confounded by changes in locomotion.

Endocannabinoids (eCBs) exert multiple effects both at cellular and behavioral levels (*Benarroch, 2014*). These molecules usually act as 'on-demand' retrograde messengers, following the activation of glutamatergic or gabaergic receptors in the postsynaptic terminal. Activation of CB1 receptors by eCBs increases the presynaptic terminal permeability to potassium while reducing calcium influx. As a result, the presynaptic terminal is hyperpolarized and the neurotransmission blunted (*Piomelli, 2003*). On the other hand, higher concentrations of eCB also act on vanilloid receptor TRPV1 (*Starowicz, Nigam & Di Marzo, 2007*). Given that TRPV1 is a ligand-gated calcium channel, its activation by eCB increases the calcium permeability in the presynaptic terminal, therefore counteracting the activation of CB1 (*Starowicz, Nigam & Di Marzo, 2007*). Indeed, the behavioral data available indicates that in most cases eCB exhibit a bell-shaped curve, with lower concentrations acting mainly through CB1 and higher quantities exerting the opposite or null effect by also acting through TRPV1 (*Fogaça et al., 2012*).

Previous studies indicate that, upon activation by AEA or 2AG, CB1 can couple to TRKB, and triggers its activity, underlying the maturation of cortical neurons (*Berghuis et al., 2005*). The observed CB1:TRKB coupling did not involve the increase of BDNF production, both at translation or transcription level. In line with these findings, we observed that AEA administration in cultured cortical cells of rat embryo was indeed able to increase the levels of pTRK in a CB1-dependent manner. To verify other possible aspects of CB1:TRKB interaction, we tested if activation of TRKB by BDNF requires previous activation of CB1 or it is, at least partially, dependent on eCB production. Thereby, we found that TRKB activation by BDNF in our experimental conditions (cultivated embryonic cortical cells) did not demand any level of CB1 activation. On the other hand, in preparations of cerebellar granule neurons, the activation of TRKB by BDNF increased the neuronal sensitivity to eCB by increasing CB1 and decreasing the

expression of the degrading enzyme monoacylglycerol lipase (*Maison et al., 2009*). Other studies, using preparations of somatosensory cortex, suggest that GABAergic synapses are inhibited by a TRKB/PLCgamma1-dependent production of eCB from the postsynaptic pyramidal cell (*Lemtiri-Chlieh & Levine, 2010*; *Zhao & Levine, 2014*; *Zhao, Yeh & Levine, 2015*). Finally, the probability of glutamate release in somatosensory cortex is decreased by BDNF/TRKB-induced eCB release (*Yeh, Selvam & Levine, 2017*); and eCBs are suggested to be produced and released to act also on midbrain dopamine neurons via previous activation of TRKB/PLC-gamma1 signaling (*Zhong et al., 2015*). Moreover, in organotypic hippocampal slice cultures, BDNF is a key downstream intermediary in CB1 protection against kainic acid-dependent excitotoxicity (*Khaspekov et al., 2004*). Thus, as previously described, TRKB activity modulates eCBs levels and the sensitivity of CB1 in response to eCB binding. Then, both eCBs and BDNF can modulate each other but, concerning the direct activation of TRKB, a 'hierarchical' relationship between these two receptors is established, with CB1 activation triggering phosphorylation of TRKB.

The higher concentration of AEA was more effective to increase the levels of pTRK through a TRPV1-dependent mechanism. Confirming this observation, capsaicin (CPS) administration also increased pTRK levels in the same system. Our data suggest that AEA acts through TRPV1 receptors to release BDNF and increase pTRK levels. Interestingly, the levels of pTRK were decreased from 325% to 200% above the control levels (same level reached with AEA 100nM) in cells treated with AEA 200 nM following the TRPV1 antagonist capsazepine or TRKB.Fc. Accordingly, the effect of higher concentration of AEA (200 nM) on pTRKB levels was only mildly attenuated by AM251. Thus, it is plausible that lower concentrations of eCB could induce modest activation of TRKB via CB1 coupling whereas higher concentrations, acting in both receptors CB1 and TRPV1, would boost BDNF release and pTRK levels. However, it is important to consider that CB1 also acts as a Gq-coupled receptor, although this mechanism is mostly found in glial cells (for review see *Oliveira da Cruz et al., 2016*), and these cells uptake rather than release BDNF (for example (*Stahlberg, Kuegler & Dean, 2018*). Therefore a CB1-mediated release of BDNF upon calcium influx is an unlikely scenario for the activation of TRKB.

Next, we used the marble burying test (MBT) to further investigate *in vivo* the relevancy of BDNF/TRK modulation by CB1 and TRPV1. This test was initially proposed as a model of neophobic anxiety (*Njung'e & Handley, 1991*), being sensitive to a variety of antidepressants, anxiolytics and antipsychotics (*Broekkamp et al., 1986*; *Bruins Slot et al., 2008*). However, this proposal has been questioned by recent studies (*Thomas et al., 2009*). The burying behavior seems to reflect natural motor programs presented in the CSTC circuitry, rather than a novelty-induced anxious state. In accordance with prior data described by our group (*Gomes et al., 2011*), mice systemically treated with the CB1 agonist WIN display a reduced burying behavior. Such effect of WIN was counteracted by the previous administration of AM251 (a CB1 antagonist); and, in the present study, by the TRK blocker k252a. Therefore, as suggested by *in vitro* results, *in vivo* data supports BDNF/TRKB signaling as a downstream trigger of CB1-induced behavioral changes.

In this scenario, given that the PFC is one of the major hubs of CSTC, we tested the effects of CB1, TRKB and TRPV1 activation in this structure. Our data indicates that
ACEA administration into PFC decreases the number of buried marbles. Surprisingly, the infusion of BDNF into PL-PFC increased the number of marbles buried. A similar facilitatory effect was also observed with CPS, and both ACEA and CPS effects were blocked by previous administration of k252a. Opposed behavioral effects are usually observed when, after increasing concentrations of AEA, there is also a targeting of eCBs to TRPV1. For example, whereas CB1 is usually responsible for the anxiolytic-like effect of low AEA doses, the activation of TRPV1 signaling by higher doses is associated with null or even anxiogenic-like effect (*Casarotto, de Bortoli & Zangrossi Jr, 2012*; *Fogaça et al., 2012*). Thus, our behavioral results suggest that in the PL-PFC BDNF and TRPV1 play a similar role.

Taken together, our results indicate that in the PL-PFC, upon activation CB1 couples to TRKB in a BDNF-independent manner (as suggested by Berghuis and colleagues) to induce a modest increase of pTRK levels and an anticompulsive-like effect. TRPV1 activation by high AEA concentrations further boosts BDNF release counteracting the CB1-mediated effect. Exogenous BDNF administration, by producing high local concentrations of this growth factor, would mimic the effects observed after TRPV1 activation. One possible explanation, as depicted in Fig. 6, for these results would be the existence of two distinct populations of neurons in the PL-PFC that colocalize TRKB:CB1 and TRKB:TRPV1, modulating the direct and indirect pathways of the CSTC circuitry, respectively. The second possibility is that the CB1- and TRPV1-dependent activation of TRKB occurs in pre and postsynaptic neurons, respectively. In this scenario, CB1 and TRPV1 would play opposed effects in the same pathway of the CSTC. These two scenarios, however, are not mutually exclusive. In agreement with this proposal, we observed a low level, although non-negligible, of co-immunoprecipitation between TRPV1 and TRKB or CB1. TRPV1, CB1, and TRKB immunolabeling in the PL-PFC slices indicate a partial overlap of these three receptors. The majority of the cells in the PL-PFC slices were double-positive to TRPV1/CaMK2, while CB1 surrounds CaMK2+cells.

In addition, both CB1 and TRKB are mutually expressed in cortical layers 2/3 and 5 (*Yeh, Selvam & Levine, 2017*). CB1 and TRPV1 receptors are also found in different neuronal populations. For instance, Marsicano and Lutz (*Marsicano & Lutz, 1999*) observed that nearly all CB1 positive cells in multiple cortical layers are also GAD65+or CCK+, suggesting a GABAergic phenotype. On the other hand, TRPV1 is present in hippocampal pyramidal neurons (*Cristino et al., 2006*). However, some studies point also to the occurrence of co-localization between both CB1 and TRPV1 receptors, thus indicating a more complex interaction (*Casarotto, de Bortoli & Zangrossi Jr, 2012*; *Fogaça et al., 2012*).

## CONCLUSION

Our *in vitro* and *in vivo* data suggest the presence of a complex interaction between CB1, TRPV1, and TRKB culminating in a final balance between BDNF-dependent and independent signaling to induce or inhibit burying behavior, respectively. Regulation of the amount of BDNF released in the synaptic cleft would be an additional mechanism, beyond inherent receptor properties, by which CB1 and TRPV1 activation usually produce opposite effects. Therefore, endocannabinoids acting upon CB1 and TRPV1, could regulate

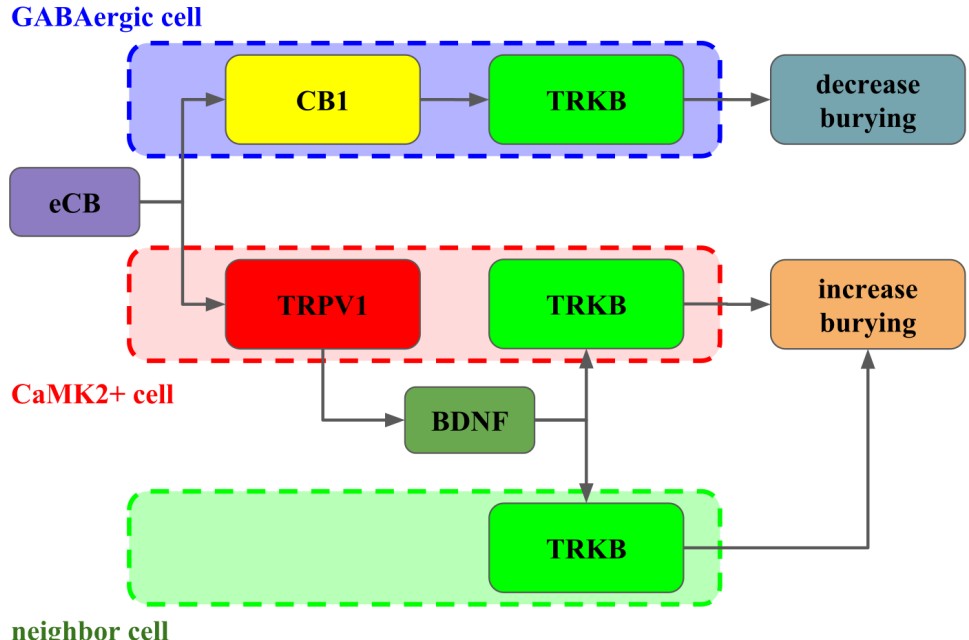

**GABAergic cell**

**CaMK2+ cell**

**neighbor cell**

**Figure 6  Interaction between CB1, TRPV1 and TRKB regulating burying behavior.** The activation of CB1 by endocannabinoids (eCB) leads to TRKB coupling, activation and decrease in burying, putatively in GABAergic cells (dashed blue). On the other hand, the activation of TRPV1 receptors by eCB triggers the release of BDNF and activation of TRKB, possibly in pyramidal (dashed red) and/or neighbor cells (dashed green) leading to increase in burying.

the TRKB-dependent plasticity in both pre- and postsynaptic compartments of different neuronal populations in a coordinated manner. It is noteworthy to highlight that *in vitro* approach, although it uses developmental brain tissue, is a powerful and useful tool to investigate the preliminary downstream signaling triggered from drug's action. It is actually difficult to transpose data from *in vitro* to *in vivo* outcomes, since important differences are found between developmental and mature tissues. However *in vitro* data as preliminary could work as a support to the *in vivo* behaviors. Thus, our behavioral data suggest that the interaction between cannabinoids and TRKB signaling, as previously observed in developmental brain tissue, might play a role in adult system. Considering the mentioned limitations, more studies are required to make our final hypothetical construct more reliable.

## ACKNOWLEDGEMENTS

We thank Flávia Salata (USP), Senem Merve Fred (UH), and Sulo Kolehmainen (UH) for their technical support.

### Funding

This work was supported by CNPq (#471382/2011-6) and FAPESP (#2011/02746-4, # 2013/01029-2, #2013/02549-0 and #2018/04250-5) for experiments conducted in Brazil, and by ERC (#322742) for experiments conducted in Finland. The funders had no role in study design, data collection and analysis, decision to publish, or preparation of the manuscript.

### Grant Disclosures

The following grant information was disclosed by the authors:
CNPq: #471382/2011-6.
FAPESP: #2011/02746-4, #2013/01029-2, #2013/02549-0 and #2018/04250-5.
ERC: #322742.

### Competing Interests

The authors declare there are no competing interests.

### Author Contributions

- Cassiano R.A.F. Diniz conceived and designed the experiments, performed the experiments, prepared figures and/or tables, approved the final draft.
- Caroline Biojone conceived and designed the experiments, performed the experiments, prepared figures and/or tables, authored or reviewed drafts of the paper, approved the final draft.
- Samia R.L. Joca and Eero Castrén contributed reagents/materials/analysis tools, approved the final draft.
- Tomi Rantamäki and Francisco S. Guimarães contributed reagents/materials/analysis tools, authored or reviewed drafts of the paper, approved the final draft.
- Plinio C. Casarotto conceived and designed the experiments, performed the experiments, analyzed the data, prepared figures and/or tables, approved the final draft.

### Animal Ethics

The following information was supplied relating to ethical approvals (i.e., approving body and any reference numbers):

All procedures were approved by the University of São Paulo (protocol: 146/2009) and for the experiments conducted at the University of Helsinki (protocol: ESAVI/10300/04.10.07/2016), in accordance with international guidelines for animal experimentation.

### Data Availability

Casarotto, Plinio (2019): Complete experimental data. figshare. Fileset. https://doi.org/10.6084/m9.figshare.6292901.v11.

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
