# Peer review of "Dual mechanism of TRKB activation by anandamide through CB1 and TRPV1 receptors"

_PeerJ, doi:10.7717/peerj.6493_

## Round 0.1 · original submission · Major Revisions

Dear Plinio,

your manuscript has been reviewed by two experts that recommended major revision. Note that both reviewers have concerns about the in vitro concentrations used. Explanation about the exclusion of some animals is also needed. I realize that extensive revision are necessary and the inclusion of new in vitro data seems imperative.

Reviewer 1 ·

Basic reporting

Language is OK.
Literature cited does not cover early work on TRPV1, or the well conducted studies that show expression of the TRPV1 is low or absent in most brain areas. There is a bit of a disconnect between differnt studies of TRPV1 expression in brain, and given the to inherent problems of just using antibodies to define protein expression, this should be acknowledged, with not just mention of supportive studies.
Figures are OK.

Experimental design

Yes.
Yes, although the "knowledge gap" is invented, not real, and given all the data is obtained by adding drugs, everything may simply be a consequence of adding drugs, not a reflection of the underlying biological reality that normally operates.
Please see below for shortcomings (only the immuno really fails).
Probably.

Validity of the findings

OK

Additional comments

This paper explores the effects of the endocannabinoid anandamide on the activation of TRK receptors in cultures of embryonic rat cortical neurons, and then attempts to use the results to interpret the effects of administration of cannabinoid and vanilloid agonists on mouse behaviour in vivo.

The key in vitro experiments consist of anandamide and capsaicin concentration response curves for promotion of the phosphorylation of TRK. The curves are incomplete, with the investigators not testing concentrations of drug above the first 1 or 2 that produce any increase in pTRK. The CRC in response to BDNF indicate that it is highly unlikely that either drug has reached anywhere near its EC50 or maximum, which means that only a fraction of the possible effect is being tested. The authors show that a CB1 antagonist blocks the effects of 100 nM AEA, implying a CB1-dependent mechanism, but they do not test the effects of the drug against 200 nM AEA, to see whether CB1 is involved in the residual response in the presence of capsazepine. The authors should also make it clear what concentration of capsazepine was used – sometimes it is written as 20 nM (probably too low be believably effective), sometimes it is written as 20 µM (probably too high to be selective). More selective ligands for CB1 and TRPV1 are available, it would re-assuring to see that they mimicked or blocked the CB1- or TRPV1-dependent effects as expected. The authors should also comment as to why they chose AEA as their endocannabinoid – in the brain the major cannabinoid transmitter is probably 2-AG, which has higher CB-receptor efficacy that AEA, and is generally found at much higher levels.

The experiments illustrated in Figure 2F are meaningless without knowing whether pTRKB levels are correlated with those of BDNF without any drug treatment, or after antagonist treatment (with either and both TRPV1 and CB1 antagonists).

The in vivo experiments utilize single concentrations of agonist (with no indication of where they lie on dose effect curves), this makes it it is hard to interpret them. It would be nice to know at minimum whether K252a blocked the effects of BDNF, whether local WIN55212 mimicked the effects of ACEA (why not AEA ?), and whether any of the agonist effects were blocked by selective antagonists. The authors may have simply chosen (or found through trial and error) doses that would give them the results that tell a story.

The immunohistochemistry “experiments” do not add much to paper. Without some sort quantification they are meaningless, without markers of glutamatergic or GABAergic neurons they cannot really support the hypothesis in Figure 6. It would also be helpful to know whether the (degree of) co-localization apparently seen in brain is also found in culture. Some comment as to why both TRPV1 and TRKB appear to largely intracellular (and diffuse) would also be useful.

No mechanism underlying the effects of the drugs are tested, no possibility that there could be synergy (or interference) between the 2 pathways tested, we don’t even know whether cell to cell communication us required for the effects in vitro.

There are a number of technical issues in the Ms. Controls for the specificity of the Ab should be included, details of how areas were chosen for immunohistochemistry and information about experimental blinding should be included. The paper lacks details about how some animals were euthanased – i.e. the mothers of the embryos, the animals that were used in the behavioural experiments. Details of post-surgical care and monitoring and the criteria for withdrawing animals from experiments should also be given.

The Results section is not written particularly clearly, the authors start with statistical statements rather than a brief description of the experiment and the result, and this comes across as stilted. Stats support science, they are essential, but should not lead description of the work.

Reviewer 2 ·

Basic reporting

The present study aims to unravel the dual effects of endocannabinoids on CB1 and TRPV1 receptors through TRKB receptors phosphorylation and link this activity to behaviour. This is done both in cell culture from rat embryo cortex and in vivo. The overall approaches used are sound, and the conclusion of interest. The report is well written (though some errors in the text remain and should be fixed), and easy to follow.

The study capitalises on a dual, explained almost as opposite, effects of endocannabinoids depending on concentration via action through CB1 or TRPV1 receptors. If this happens to be true, it is of interest. Some clarifications are needed for the validity of the finding to be properly assessed.

Introduction:

1. The description of the signalling pathways related to endocannabinoid via CB1 is weak and needs to be expanded. Gi/o is one of several possible mechanisms. Others include Gq and Gs. Gq is of importance here, as they cause, as for TRPV1 channels, an increase in calcium. Some agonists, such as N-arachidonoyl-dopamine, for example, are not only Gq-specific, but also a relatively potent ligand for TRPV1. Anandamide can also increase calcium through the RE via Gq.

2. There appears to be some confusion regarding a bell-shaped effect of endocannabinoids and TRPV1, which should be better described, as two main concepts are mixed together, the first one being basic pharmacology, the second being behavioural outcome. From a pharmacology standpoint, an increase in the concentration of eCBs is not going to favour TRPV1 over CB1. An increase in eCB concentration will reach maximal efficacy on CB1 as well as start activating TRPV1 receptors (as eCBs possess a lower affinity for the channel). Furthermore, most of these effects are going to take place in different cells.

3. It is not clear from reading this introduction what mechanism exactly links TRKB to CB1 and TRPV1, in fact there is no mention of TRKB to TRPV1 whatsoever. Is this interaction already known or part of the results of the current study? This is of great importance as not only is this paper stating that they are part of the same type of signalling mechanisms, but the authors claim that there is protein-protein interaction as confirmed from co-immunoprecipitation. The localization (pre-post synaptic) of these receptors need to be better determined as well.


Methods:

1. In experimental design, line 211, “dose-response effects…”should be “concentration-response effects”. Please fix other mentions of in vitro work as dose-response throughout the paper.

2. Line 218-220: the 2 sentences are confusing. First one is AM251 followed by AEA. Second one is AM251 but NOT followed by BDNF? Please rephrase.

3. Overall please use µ instead of u. It could be good to use inverse agonist rather than antagonist as it is a more proper pharmacology term.

4. The concentration/doses for local drug administration are very low overall. Most are below the known concentration for displacement of radio-ligand in cell-culture based assays. It would be judicious to comment on the reason for the choice of concentration.

5. The reason why this study mostly focuses on endocannabinoids, but then use synthetic ones for the burying test needs to be assessed.



Discussion:

1. Line 347 forward: I am not certain I understand how this can be claimed. In figure 1a/b, AEA causes higher increase of pTRK compared to capsaicin. AEA is a very weak agonist to TRPV1 at best compared to capsaicin. This does not make much sense as far as pharmacology is concerned. There is nowhere in this paper where there is a clear result showing that “even if AEA has a higher affinity for CB1 than for TRPV1, the related efficacy appears to be inverted in favour of TRPV1”. I do not see this shown in this study. If anything, it looks like activity on CB1 and TRPV1 are different, and via different mechanisms as CB1 activity does not cause an increase in BDNF.

2. The in vitro work is done is cell culture. The in vivo behavioural work is done acutely. This is a major distinction as endocannabinoids are involved in neural development. This needs to be properly discussed.

Figures:

The images and figures need a serious rework. Harmonisation of fonts, as well as higher quality of images are required. In its current state, they do not look ready for publication.

Experimental design

The overall experimental design is of interest, and original. The research questions are well defined and relevant, though they might be based on somewhat weak foundation (the bell shape effect of endocannabinoids on CB1 vs TRPV1). The technical level is overall good, though the immunohistochemistry images are not currently convincing.

1. Is the burying done in a clean box? This is a major stressor for mice and could be important for marble burying behaviour, especially when injecting cannabinoids which could further induce anxiety-like behaviours.

2. If high doses of endocannabinoids, as hypothesized, act through TRPV1, then one would think that high concentration of endocannabinoids would increase burying behaviour, especially when localised injections are involved (rather than systemic). Why was this not tested?

3. As mentioned previously, overall concentration used for endocannabinoids are low, and concentration response curve for pTRK levels incomplete. Please explain if higher concentrations were also tested.

4. Why were synthetic cannabinoids used for the burying test rather than endocannabinoids?

5. It would be good to provide lower magnification of the slices to have an overall idea of the PFC structure. Layer information would also be relevant.

Validity of the findings

Data is mostly robust, though some confirmations are needed for this manuscript to be accepted.

1. Result from 1d does not look like a very convincing coIP. This is not helped by the omission of any statistical analysis, or sample size.

2. If one does not look at the scale of the y-axis in 1a/b, one might be inclined to think that the effect of AEA and CPS on pTRK levels are very similar. % change from baseline should be mentioned in the text. AEA causes higher pTRK compared to capsaicin, yet one of the basis of this study is that AEA, a weak TRPV1 agonist, causes increased burying at higher doses. This should be discussed more properly, as it is a foundation of the paper. As well, none of the concentration response curve are complete. Were higher concentration tested?

3. For 2a, did AM251 decrease basal pTRK levels? This is important to point out as it could hint at tonic activity.

4. I can see from the provided Prism data that AEA was conducted for the burying test, but it was omitted from this report. This is quite surprising as this is an endocannabinoid paper. Justification for this choice is essential.

5. I can also see from the Prism data that the two only excluded datapoints from the burying data are 2 zeros from the ACEA + 252a group. If these were included, there would be no significant rescue. If anything, the 9 marbles buried from this group is in fact the outlier. Justification for this omission is primordial for this study to be published. Same goes for the 9 marbles as possible outlier for the CPS + 252a group.

6. For the immunohistochemistry images, it is not too clear what the white arrows are pointing at. Background levels are quite high for CaMK2. Were blocking peptides used as control? There is a lot of TRPV1 staining in the soma rather than the cell surface, which is not what one would expect from the literature. If this is PFC slices, please provide lower magnification for an overview of the structure as a whole. Better controls are required to assess the validity of the immunohistochemistry overall.

7. Best practice for the use of colours for immunohistochemistry should be used. Arrows are not clearly showing what they should. Colour scheme makes colocalization dubious. Green/red is particularly to be avoided for colour-blind people.

8. The schematic states that CB1 is on GABAergic neurons but this claim is not-substantiated in any way in this manuscript.. Is this to be taken from the literature? CB1 are on both excitatory and inhibitory neurons. Neighbour cells is also very vague term.

Additional comments

It is of great importance to explain why two values were omitted from the burying test for ACEA + 252a group. The CP+k252a group is not too convincing either as the 9 marbles buried for one animal could potentially be an outlier. This is of great importance for the overall conclusion and validity of this study.

---

## Round 0.2 · accepted · Accept

I have looked at the rebuttal letter and tracked changes manuscript and I realize that the major concerns of the reviewers were addressed adequately.

#